# Evaluation of blood cell count parameters as predictors of treatment failure of malaria in Angola: An observational study

**Euclides Nenga Manuel Sacomboio**[1,2,3]*, **Cruz dos Santos Sebastião**[1,2,4], **Silvana Teresa da Costa Salvador**[2], **Joaquim António João**[2], **Daisy Viviana Sebastião Bapolo**[5], **Ngiambudulu M. Francisco**[1], **Joana Morais**[1,6], **Eduardo Ekundi Valentim**[7]

**1** Instituto Nacional de Investigação em Saúde (INIS), Luanda, Angola, **2** Instituto Superior de Ciências de Saúde (ISCISA), Universidade Agostinho Neto (UAN), Luanda, Angola, **3** Centro de Investigação em Saúde de Angola (CISA), Caxito, Angola, **4** Instituto Superior de Ciências de Saúde (ISCISA), Universidade Agostinho Neto (UAN), Luanda, Angola, **5** Anglia Ruskin University (ARU), Chelmsford/England, United Kingdom, **6** Faculdade de Medicina, Universidade Agostinho Neto, Luanda, Angola, **7** Instituto Politécnico, Universidade Rainha Nginga a Mbande (URNM), Malanje, Angola

* euclides.sacomboio@ucan.edu

**Data Availability Statement:** All relevant data are within the paper and its Supporting information files.

## Abstract

### Background

Despite the guidelines provided by the World Health Organization for the treatment of malaria, treatment failure occurs in many hospitalized patients.

### Objective

Evaluate whether blood cell count parameters may serve as predictors for malaria treatment.

### Methodology

A cross-sectional study with a quantitative approach.

### Results

Of the 219 patients, 21.5% showed failure to antimalarial treatment, Patient with 21 and 40 years (72.6%), male (53.4%), from peri-urban area (47.5%), with high parasitemia (59.8%), treated with Arthemeter (90.9%) and the mortality were 5.9%. Significant associations were observed between occupation, level of parasitemia and outcome with resistance to antimalarial treatment (p<0.05). Patients with normal Hb [OR: 0.75 (95% CI: 0.39–1.44), p = 0.393], RBC [OR: 0.83 (95% CI: 0.40–1.72), p = 0.632], RDW [OR: 0.54 (95% CI: 0.27–1.09), p = 0.088], MCV [OR: 0.61 (95% CI: 0.28–1.31), p = 0.204] were less likely to have malaria treatment failures after artemisinin-based therapy failure. In contrast, those with normal values of segmented neutrophils [OR: 0.32 (95% CI: 0.11–0.96), p = 0.042] and lymphocyte counts [OR: 0.24 (95% CI: 0.05–1.04), p = 0.055]. We also found that patients with significant low levels of Hct [OR: 0.31 (95% CI: 0.15–0.64) p = 0.002], and high leukocytes

**Funding:** Euclides Sacomboio. Grant. 2018.2 International Society of Neprhology.

**Competing interests:** The authors have declared that no competing interests exist.

[OR: 8.88 (95% CI: 2.02–37.2), p = 0.004] and normal platelet values [OR: 1.42 (95% CI: 0.73–2.95), p = 0.280] demonstrated high probability of treatment failure.

## Conclusion

The importance of blood cell count parameters in monitoring malaria therapy necessitates the urgent need to re-evaluate Artemether-based therapy. Future studies involving more participants in different settings are needed to provide further evidence.

## Background

Globally, there were an estimated 229 million malaria cases in 2019 in 87 malaria-endemic countries, although there is a decline compared to the year 2000 [1], in Angola, a record of 3,7 million new cases was registered during the first 5 months of 2021, and 5,573 deaths were reported in the same period [1, 2]. The current figures for disease, morbidity, and mortality may differ from the figures reported above due to the weak capacity of surveillance, diagnosis, and underreporting of cases [2].

Studies have reported that hematological changes are among the most common complications found in malaria, and the prediction of hematological changes allows the clinician to establish an effective and early therapeutic intervention to prevent the occurrence of major complications since these parameters are measurable indicators of blood components that serve as disease diagnostic biomarkers [3, 4].

Hematological abnormalities are considered a hallmark of malaria. A study has that many of these hematological values can lead to an increased clinical prediction of malaria, suggesting for immediate initiation of specific therapy, even in the absence of a positive smear report for malaria [5]. Previous studies have reported episodes of malaria accompanied by an anemic condition, thrombocytopenia, splenomegaly, mild to moderate atypical lymphocytosis, and cases of disseminated intravascular coagulation can rarely occur, as well as there are reports of the occurrence of leukopenia and leukocytosis [6, 7].

The clinical manifestation of *Plasmodium falciparum* malaria ranges from mild acute febrile illness to severe, life-threatening systemic complications involving multiple organ failure, which can cause significant hematologic changes ranging from hemolytic anemia (Hb<10 g/dl.), defective erythropoiesis and reticulocyte production, leukocytosis (WBC>11 X 103/Microliter), leukocytopenia (WBC<4 X103/Microliter), thrombocytopenia (TCP) platelet count less than 150 x103/Microliter, platelet dysfunction in severe malaria and intravascular coagulation disseminated disease (CID) [8, 9].

Several studies try to show how malaria infection affects the blood count, in one of them it was found that the hematology of malaria infection was characterized by cytopenia or pancytopenia (P < 0.001, OR = 11.14) of which thrombocytopenia was the most prominent component (P< 0.001, OR = 37.94), followed by anemia due to a reduction in red blood cell count (RBC) affecting a decrease in red blood cell volume (PCV) (P < 0.003, OR = 2.13) with a high red blood cell distribution width (P < 0.025, OR = 1.78), where higher parasite density was associated with increased incidence of anemia and severe thrombocytopenia, PDW was elevated (P < 0.001, OR = 6.93) and PCT was reduced (P < 0.001, OR = 123.64) in positive cases, which led the authors to conclude that thrombocytopenia with reduced PCV or reduced erythrocyte count is a hallmark of malaria [10].

Also, Kotepui and colleagues, (2014) demonstrated that patients infected with malaria had significantly reduced red blood cell count (RBCs), hemoglobin (Hb), platelet count, white

blood cell count (leukocyte), neutrophils, monocytes, lymphocytes, and eosinophils, while corpuscular mean volume (MCV), corpuscular mean hemoglobin (MCH), corpuscular mean hemoglobin concentration (MCHC), neutrophil-lymphocyte ratio -(NLR) and monocyte-lymphocyte ratio (MLR) were higher compared to patients not infected with malaria [11].

The use of artemisinin and its derivatives is becoming increasingly important in the treatment of malaria, however, it is necessary to use it in combination with second-line antimalarial drugs to increase efficacy and avoid treatment failure, which may lead to the development of drug resistance [12, 13]. Antimalarial drugs such as mefloquine, halofantrine, quinine, and lumefantrine are relatively weak inhibitors of β-hematin formation in vitro, thus the molecular basis of their antimalarial activity is far from clear and may involve additional targets to the hemoglobin degradation/heme detoxification pathway [13–15].

Although increasing scientific evidence has shown hematological alterations resulting from malaria, little evidence exists on whether alterations of blood cell count may predict malaria treatment failure that may lead to possible malaria drug resistance. The present study aimed to evaluate blood cell count parameters in the prediction of treatment failure in patients hospitalized for malaria.

## Methodology

### Study design and ethical statement

The study was conducted as a cross-sectional study with a quantitative approach. It was approved by the Human Research Ethics Committee of the Higher Institute of Health Sciences (No. 755/GD/ISCISA/UAN/2018) and by the clinical directorate of Hospital Josina Machel—Maria Pia in Luanda (No. 260/DPC/HJM/2018). In addition to the evaluation of the research project, the Ethics and Research Committee involving human beings also approved the free and informed consent form that is given to the patient to be signed in two copies (one for the researcher and one for the patient), the free consent term contains all information about the project, including the name, telephone number and email of the principal investigator, in cases where the patient withdraws from the possibility of being part of the study. All patients who agreed to participate in this study had to sign an informed consent form after being informed about the nature and objectives of the study through the consent form and clarifying all doubts.

### Patient recruitment

The study population consisted of 219 out of 410 patients admitted and hospitalized for malaria complications at the Hospital Josina Machel—Maria Pia, in Luanda, Angola, between December 2018 and January 2020. A 95% confidence index was maintained with a margin of error of 5%. Only patients who have met the selection criteria and agreed to participate in the study were included. Additional information was collected through an open and closed questionnaire for patients aged 12 to 66 years old and only patients who were hospitalized for more than 5 days were included. Only patients who confirmed that they had not received any pre-hospital treatment for malaria were included in the study. All patients with a history of hypertensive disease, diabetes, chronic kidney disease, cerebral malaria, other chronic diseases, immunocompromised, and other pathologies contributing to treatment failure were excluded to avoid bias in data analysis.

### Laboratory procedure

The erythrogram and white blood cell data were evaluated on the admission of patients before starting antimalarial treatment. For the erythrogram data, hemoglobin (Hb), red blood cell count

(RBCs), Red Cell Distribution Width (RDW), hematocrit (Hct), mean corpuscular volume (MCV) and mean corpuscular hemoglobin concentration (MCHC) were evaluated. For the Leukogram data, the lymphocytes, platelets, neutrophils, and leukocytes counts were evaluated [16]. The study did not include mean corpuscular hemoglobin (MCH), monocyte count, eosinophil count, neutrophil-to-lymphocyte ratio (NLR), and monocyte-to-lymphocyte ratio (MLR) due to the devices that had some problems in reading these exams in some of the patients, as we did not obtain data from all patients included in the study, these data were not analyzed in the study because they were not complete for all patients. In cases where the reference values of blood cell count components were different between men and women, the reference values were adjusted for the whole group based on the minimum value for women and the maximum value for men, in addition, it was performed adjustment according to the references according to the specific characteristics of the Angolan population, as can be seen in the presentation of the results.

A malaria diagnosis screening test was performed using rapid malaria antigen tests (SD-Bioline Malaria AG Pf/PAN), followed was confirmed by microscopy technique of direct visualization of the parasite by Giemsa-stained peripheral blood thickening [17]. Patients who presented parasitemia less than or equal to 1000 p/mm3 were classified as moderate parasitemia whilst patients who presented parasitemia above 1000 p/mm3 were classified as high parasitemia [14, 15]. We confirmed malaria treatment failure by evaluating the parasitemia of patients 5 days after finishing the treatment with antimalarial drugs (Arthemeter or Artesunate), which have already been shown in other studies that under normal conditions can reduce parasitemia from 100,000 p/mm3 to undetectable parasitemia after 5 days of in-hospital treatment [17–19]. Patients who did not have a complete reduction in parasitemia after 5 days of treatment were classified to have treatment failure and those who had a complete reduction in parasitemia were considered to be cured.

Although it is known that nutritional status, the prevalence of intestinal parasites, other pathological conditions, and sociodemographic data can influence the hematological indices of patients residing in malaria-endemic areas, as there is no clinical information in the patients' files about the aforementioned situations, we believe that the presence of confounding factors was reduced as these patients were treated by the medical team and submitted to different types of clinical examinations.

## Statistical analysis

Data obtained was categorized and analyzed using IBM® SPSS Statistics v25. Absolute and relative frequencies were determined. The chi-square (X2) and logistic regression tests were used to assess the relationship between categorical variables. The odds ratio (OR) and its 95% confidence intervals (CI) were calculated to assess the strength and direction of the relationship. All reported p-values are two-tailed and deemed significant when p<0.05. Pearson's Chi-Square test is generally used to compare two categorical variables and check whether they are homogeneous with each other and allowed us to compare the phenomena we studied in relation to treatment failure (with failure/without failure) and the Odds Ratios was used to verify the strength of association between different phenomena that we studied and the presence or absence of treatment failure.

## Results

### Sociodemographic and clinical characteristics related to antimalarial treatment failure

The sociodemographic and clinical characteristics related to antimalarial treatment failures are summarized in Table 1. Of the 219 patients who were included in the study, 21.5% (n = 47/

**Table 1. Characteristics sociodemographic and clinical of patients with antimalarial resentence treatment in Luanda, Angola.**

| Characteristics | | Resistance | | Chi-Square |
|---|---|---|---|---|
| | N (%) | No (%) | Yes (%) | p-value |
| **Age** | **219(100)** | **172 (78.5)** | **47 (21.5)** | |
| ≤20 years | 40 (18.3) | 32 (80.0) | 8 (20.0) | 0.151 |
| **21–40 years** | 159 (72.6) | 121 (76.1) | 38 (23.9) | |
| ≥ 41 years | 20 (9.1) | 19 (95.0) | 1 (5.0) | |
| **Gender** | | | | |
| Female | 102 (46.6) | 80 (78.4) | 22 (21.6) | 0.945 |
| Male | 117 (53.4) | 92 (78.6) | 25 (21.4) | |
| **Occupation** | | | | |
| Unoccupied | 56 (25.6) | 56 (100.0) | 0 (0.0) | **<0.001** |
| Students | 64 (29.2) | 46 (71.9) | 18 (28.1) | |
| Freelance | 39 (17.8) | 30 (76.9) | 9 (23.1) | |
| Appointed | 60 (27.4) | 40 (66.7) | 20 (33.3) | |
| **Residence area** | | | | |
| Urban | 66 (30.1) | 54 (81.8) | 14 (18.2) | 0.165 |
| Peri-urban | 104 (47.5) | 77 (74.0) | 27 (30.0) | |
| Rural | 49 (22.4) | 43 (87.8) | 6 (12.2) | |
| **Parasitemic degree** | | | | |
| Moderate | 53 (24.2) | 28 (52.8) | 25 (47.2) | **<0.0001** |
| High | 131 (59.8) | 116 (88.5) | 15 (11.5) | |
| Hyper | 35 (16.0) | 28 (80.0) | 7 (20.0) | |
| **Antimalarial treatment** | | | | |
| Artemether | 199 (90.9) | 156 (78.4) | 43 (21.6) | 0.876 |
| Artesunate | 20 (9.1) | 16 (80.0) | 4 (20.0) | |
| **Outcomes** | | | | |
| Discharged | 72 (32.9) | 50 (69.4) | 22 (30.6) | **0.047** |
| Keep | 134 (61.2) | 110 (82.1) | 24 (17.9) | |
| Dead | 13 (5.9) | 12 (92.3) | 1 (7.7) | |

219) exhibited a failure to antimalarial treatment. In contrast, 78.5% (n = 172/219) were successful cured antimalarial treatment. All patients lived in Luanda province and had been infected with *Plasmodium falciparum* (P. falciparum). Patients with ages ranging from 21 to 40 years old represented approximately 72.6% (n = 159/219) of patients studied male patients represented 53.4% of the studied population (n = 117/219), 29.2% of patients were students (n = 64/219), patients from the peri-urban areas of Luanda province represented 47.5% (n = 104/219). Patients with high parasitemia represented 59.8% (n = 131/219), and 90.9% of them were treated with Arthemeter, while patients' mortality in this study was 5.9% (n = 13/219). There was a significant relationship between occupation, level of parasitemia, and clinical outcome with resistance to antimalarial treatment (p<0.05).

## Erytrogram and antimalarials treatment failure among patients with malaria in Luanda, Angola

We evaluated whether the erythrogram count on hospitalized patients under antimalarial therapy could predict failure of treatment (Table 2). We found that 59.9% (n = 129/219) of patients with malaria had Hb values below (mean 9.4 mg/dL, SD = 1.4), the reference values used in our Laboratory (11 to 19 mg/dL), 32.4% (71/219) of patients had RBC below (mean 3.1

**Table 2. Erytrogram and antimalaric resistance among patients with malaria in Luanda, Angola.**

| Characteristic | N (%) | Resistance | | | Univariate analysis | | Multivariate analysis | |
|---|---|---|---|---|---|---|---|---|
| | | No (%) | Yes (%) | p-value | OR (95% CI) | p-value | OR (95% CI) | p-value |
| Overall | 219 (100) | 172 (78.5) | 47 (21.5) | | | | | |
| **Hemoglobin (mg/DL)** | | | | | | | | |
| Low | 129 (59.9) | 104 (80.6) | 25 (19.4) | 0.393 | 1 | - | 1 | - |
| Normal | 90 (41.1) | 68 (75.6) | 22 (24.4) | | 0.75 (0.39–1.44) | 0.393 | 0.75 (0.39–1.44) | 0.393 |
| **RBC (million/μL)** | | | | | | | | |
| Low | 71 (32.4) | 57 (80.3) | 14 (19.7) | 0.895 | 1 | - | 1 | - |
| Normal | 148 (67.6) | 115 (77.7) | 33 (22.3) | | 0.83 (0.40–1.72) | 0.632 | 0.83(0.41–1.72) | 0.835 |
| **RDW (%)** | | | | | | | | |
| Low | 165 (75.3) | 134 (81.7) | 31 (18.3) | 0.086 | 1 | - | 1 | - |
| Normal | 54 (24.7) | 38 (70.4) | 16 (29.6) | | 0.54 (0.27–1.09) | 0.088 | 0.54 (0.26–1.09) | 0.088 |
| **Hematocrits** | | | | | | | | |
| Low | 102 (46.6) | 90 (88.2) | 12 (11.8) | **0.001** | 1 | - | 1 | - |
| Normal | 117 (53.4) | 82 (70.1) | 35 (29.9) | | 0.31 (0.15–0.64) | **0.002** | 0.31 (0.15–0.64) | **0.002** |
| **MCV (fL)** | | | | | | | | |
| Low | 63 (28.8) | 53 (84.1) | 10 (15.9) | 0.201 | 1 | - | 1 | - |
| Normal | 156 (71.2) | 119 (76.3) | 37 (23.7) | | 0.61 (0.28–1.31) | 0.204 | 0.60 (0.28–1.31) | 0.204 |
| **MCHC(mg/dL)** | | | | | | | | |
| Normal | 172 (78.5) | 133 (65.7) | 39 (34.3) | 0.383 | 1 | - | 1 | - |
| High | 47 (21.5) | 39 (82.9) | 8 (17.1) | | 1.45 (0.63–3.36) | 0.385 | 1.45 (0.62–3.36) | 0.385 |

Observation: Bold results mean they were significant in the chi-square or logistic regression (p<0.05)

[#]Adjusted for all the explanatory variables listed

million/μL, SD = 0.5), reference values (between 3.9 to 5.9 million/μL), 75.3%(165/219) of malaria patients had red cell distribution width (RDW) below (mean 6.9%, SD = 1.5), reference values (between 10.0 to 16.0%), 46.6% (102/219) had Hct below (mean 28.9%, SD = 5.5), the reference values (between 35 and 53%), 28.8% (63/219) of the patients had mean corpuscular volume (MCV) below (mean 68.4 fL, SD = 18.4), the reference values (between 80.0 100.0 fL) and 21.5%(47/219) presented mean corpuscular hemoglobin concentration (MCHC) above (mean 37.9 g/dL, SD = 3.5), the reference values (between 31.0 to 36.0 g/dL). We also found a statistically significant relationship between antimalarial treatment failure and the Hct count (p < 0.05), while the Hb count, RBC, RDW, MCV, and MCHC were not associated with antimalarial treatment failure (p>0.05).

In the univariate analysis, it was found that the resistance rate was higher in the group with normal hemoglobin (24.3%) and lower in the group with low hemoglobin (9.5%), although there was no significant relationship between resistance to antimalarial treatment and many components of the erythrogram, patients with normal Hb values [OR: 0.75 (95% CI: 0.39–1.44), p = 0.393], RBC [OR: 0.83 (95% CI: 0.40–1.72), p = 0.632], RDW [OR: 0.54 (IC of 95%: 0.27–1.09), p = 0.088], MCV [OR: 0.61 (95% CI: 0.28–1.31), p = 0.204] were less likely to develop resistance to treatment when compared to patients with values below the reference values. While patients with high values of MCHC [OR: 1.45 (95% CI: 0.63–3.36), p = 0.385] were more likely to present resistance to antimalarial treatment when compared to patients who presented MCHC normal. On the other hand, there was a significant relationship between resistance to antimalarial treatment and normal Hct count [OR: 0.31 (95% CI: 0.15–0.64), p = 0.002] with a low probability of resistance to antimalarial treatment, when compared to patients with Hct below reference values.

**Table 3. Leukogram and resistance to antimalarial treatment.**

| Characteristic | N (%) | Resistance | | | Univariate analysis | | Multivariate analysis | |
|---|---|---|---|---|---|---|---|---|
| | | No (%) | Yes (%) | p-value | OR (95% CI) | p-value | OR (95% CI) | p-value |
| Overall | 219 (100) | 172 (78.5) | 47 (21.5) | | | | | |
| **Lymphocyte (mg/DL)** | | | | | | | | |
| Low | 29 (13.2) | 27 (93.1) | 2 (6.9) | **0.040** | 1 | - | 1 | - |
| Normal | 190 (86.8) | 145 (76.3) | 45 (23.7) | | 0.24 (0.05–1.04) | 0.055 | 0.24 (0.05–1.04) | 0.057 |
| **Platelets (million/μL)** | | | | | | | | |
| Low | 139 (63.5) | 106 (76.3) | 33 (23.7) | 0.279 | 1 | - | 1 | - |
| Normal | 80 (36.5) | 66 (82.5) | 14 (17.5) | | 1.47 (0.73–2.95) | 0.280 | 0.68 (0.34–1.36) | 0.280 |
| **Neutrophils (%)** | | | | | | | | |
| Low | 42 (19.2) | 38 (90.5) | 4 (9.5) | **0.035** | 1 | - | 1 | - |
| Normal | 177 (80.8) | 134 (75.7) | 43 (24.3) | | 0.32 (0.11–0.96) | **0.043** | 0.32 (0.11–0.96) | **0.043** |
| **Leukocyte** | | | | | | | | |
| Normal | 170 (77.6) | 125 (73.5) | 45 (26.5) | **0.001** | 1 | - | 1 | - |
| High | 49 (22.4) | 47 (95.9) | 2 (4.9) | | 8.88 (2.02–37.2) | **0.004** | 8.37 (1.93–36.3) | **0.005** |

Observation: Bold results mean they were significant in the chi-square or logistic regression (p<0.05)

[#]Adjusted for all the explanatory variables listed

## Leukogram count analysis during antimalarial treatment in the adult Angolan hospitalized patients

To access whether the leukogram count could provide insight on the treatment course and outcome of adult hospitalized patients, we analyzed 219 blood samples from patients under antimalarial treatment (Table 3). We noticed that 13.2% (n = 29/219) of the patients with malaria had lymphocyte (LT) values below (mean 470/μL, SD = 10.5), the reference ranges (900 and 4,000/μL), 63.5% (n = 139/219) of the patients had platelets (PT) below (mean 89,000/μL, SD = 25.000), reference values (140,000 to 450,000/μL), 19.2% (n = 42/219) of patients with malaria had segmented neutrophils (NT) below (mean 630/μL, SD = 200.5), reference values (1,600 and 8,000/μL), and 77.6% (n = 170/219) had a mean concentration of total leukocytes (LC) below (mean 700/μL, SD = 200.6), the reference values (between 4,000 and 11,000/μL). The data show a statistically significant relationship between the prevalence of antimalarial treatment failure in malaria and the lymphocytes (LT) count, of segmented neutrophils (NT) and total leukocytes (LC) in the white blood cell count (p < 0.05), while the platelet count (PT) was not related to resistance to antimalarial treatment (p>0.05).

In the univariate analysis, we found that there was a significant relationship between antimalarial treatment failure and the presence of normal values of NT [OR: 0.32 (95% CI: 0.11–0.96), p = 0.042] with a low probability of presenting antimalarial treatment failure. There was also a significant relationship between resistance to antimalarial treatment and the presence of high values of total LC [OR: 8.88 (95% CI: 2.02–37.2), p = 0.004] with a high probability of presenting resistance to antimalarial treatment when compared to patients with normal total leukocytes. Although there was no significant relationship between antimalarial treatment failure and platelet counts, patients who had normal lymphocyte counts [OR: 0.24 (95% CI: 0.05–1.04), p = 0.055] had a low probability of being resistant to treatment antimalarial, when compared to patients with LT below reference values (Add value here). On the other hand, patients with normal platelet values [OR: 1.42 (95% CI: 0.73–2.95), p = 0.280] had a high probability of exhibiting antimalarial treatment failure when compared to patients with normal platelet

values. Interestingly, the multivariate analysis also predicted low chances of antimalarial treatment failure in patients with normal white blood cell counts.

## Discussion

The socio-demographic and clinical data related to age, gender, occupation, place of residence, degree of parasitemia, antimalarial treatment, and mortality obtained in this study, do not differ from data obtained in a previous study carried out by our research team in the same health institution, that assessed the uremic condition of patients with malaria, were we found that of the 184 hospitalized patients, most were men (68%) who came from municipalities of Luanda (36%) and Cazenga (30%), the majority were unemployed (35%), students (27%) and self-employed (22%), also 41% of these patients had high parasitemia and were treated with Arthemeter (83%), Artesunate (15%) and Quinine (2%), in that same study we reported a mortality rate of 8%, and a hospital discharge rate of 36% [19].

In the present study, antimalarial treatment failure occurred in about 21.5% of patients, in both antimalarial drugs used, both drugs showed resistance above 20%, and this response to treatment leads us to agree with there are studies that propose the production of antimalarials that aim to prevent the invasion of red blood cells because: (i) extracellular parasites are directly exposed to drugs found in the bloodstream, (ii) most of the parasite proteins necessary for the invasion lack human bioequivalent, and this technique would offer the possibilities of selective inhibition and (iii) blocking the invasion could immediately stop parasite multiplication in their blood stages [20]. Treatment failure is one of the main factors leading to antimicrobial drug resistance.

Recent studies in vitro and in vitro sought to assess artemisinin resistance and identified its causal genetic determinant, explored its molecular mechanism, and assessed its clinical impact, where it found that resistance to artemisinin manifests as slow clearance of the parasite in patients and increased survival of early-stage ring parasites in vitro and this phenomenon is caused by single nucleotide polymorphisms in the parasite's K13DP gene, which is associated with a positively regulated "split protein response" pathway that can antagonize pro-activity oxidant of artemisinin's and selects resistance to partner drugs that quickly lead to failure of combined treatments [21, 22]. A study from Uganda investigated the P. falciparum kelch protein helix domain mutations associated with delayed elimination of artemisinin and found that the prevalence of the 469Y and 675V mutations increased at various sites in northern Uganda (up to 23% and 41%, respectively) [23].

In the erythrogram analysis, except for hematocrits, which showed a statistically significant relationship with resistance to antimalarials, it was noticeable that individuals who had normal values in the components of the erythrogram were less likely to develop resistance to antimalarial treatment. To the best of our knowledge, this is the first report to report changes in blood cell count parameters that predict antimalarial treatment failure. White, 2018 stated that malaria infection causes hemolysis of infected and uninfected erythrocytes, as well as causing bone marrow dyserythropoiesis that compromises rapid recovery from anemia, thus, in areas of high malaria transmission, almost all infants, young children, and adults have a reduced hematocrit ou hemoglobin concentration as a result of various malaria infections [24]. Some studies have shown that reduced RBC deformability could serve as a predictor of antimalarial activity, suggesting that reduced cell deformability may serve as a rapid and sensitive biomarker to assess antimalarial drug efficacy [25, 26].

A study involved 2,024 malaria-infected cases that compared blood cell parameters found that RBC and Hb counts were significantly reduced in patients with high parasitemia compared to those with low and moderate parasitemia groups ($p < 0.05$), while MCV and MCH

were significantly reduced in patients with moderate parasitemia compared to those with low and high parasitemia groups (p < 0.05) [27]. It has been stated that clinical, epidemiological, and genome-wide association studies identified several polymorphisms in red blood cell (RBC) proteins that attenuate the pathogenesis of malaria, including well-known polymorphisms in hemoglobin, intracellular enzymes, red cell channels, red cell surface markers, and proteins that affect red cell cytoskeleton and red cell morphology, therefore, the authors believe that a better understanding of how changes in red and white blood cell physiology can impact the pathogenesis of malaria and resistance to treatment and thus reveal new strategies to fight the disease [28].

All leukogram components in this study had a statistically significant relationship with antimalarial drug failure, except for platelets, which shows that patients with normal leukocytes components were less likely to develop antimalarial treatment failure, this can be understood by the fact that these components perform immune response functions, as described in an analysis of data from 7 randomized controlled trials at 13 sites in 9 countries comparing artesunate-amodiaquine to single and combined treatments (including amodiaquine monotherapy, artesunate monotherapy, artemether-lumefantrine, artesunate and sulfadoxine-pyrimethamine, and dihydroartemisinin), the incidence of treatment-emergent adverse events for neutropenia was 11%, however, when neutrophil counts were compared between treatment groups, there were no apparent differences, but when artesunate was used, an effect on neutrophil numbers occurred [29]. Nevertheless, in the present study, patients who had low levels of neutrophils had greater chances of developing antimalarial treatment failure, which corroborates with a study that states that neutrophils can eliminate pathogens by phagocytosis; for the production of reactive oxygen species or reactive nitrogen species and other antimicrobial products; or by the formation of extracellular neutrophil traps, then individuals with low values of neutrophils would have greater chances of resistance to antimalarial treatment [30].

A study developed in Ethiopia found that the mean values of Hb, Hct, platelets, RBC, LC, and LT were significantly lower in patients with malaria when compared to individuals who tested negative for malaria, in the group individuals with malaria; they found that the prevalence of thrombocytopenia and anemia were 84% and 67%, respectively [21]. In a study carried out in Thailand, leukocyte and neutrophil counts were found to be significantly higher in patients with high parasitemia compared to those with low and moderate parasitemia (p<0.0001), while lymphocyte, platelets, and monocyte counts were significantly lower in patients with high parasitemia compared to those with low and moderate parasitemia groups (p < 0.0001) [27].

Hematological aspects constitute a very interesting area in malaria since hematological alterations such as anemia, thrombocytopenia and leukopenia showed a statistically significant correlation with *Plasmodium* infection, in addition to routine laboratory findings such as hemoglobin, leukocyte, and platelet counts, and even erythrocyte distribution width values can provide a diagnostic clue in a patient with acute febrile illness in endemic areas, thus increasing the likelihood of correctly diagnosing malaria and improving prompt initiation of treatment [5].

Although our study has strength in the sense that this is the first study to extrapolate the blood cell count parameters with the antimalarial treatment failure setting in a high endemic country but also in a low-income setting country. However, there are some limitations in this study: First, we did not use the molecular method to diagnose our patients. Second, eosinophils are the first line defense cells in the response against *Plasmodium*, but we were unable to analyze these types of cells due to the lack of this parameter in the technology system that we used. Third, though antimalarial drug resistance treatment may lead to treatment failure or vice versa, other factors such as is failure to clear malaria parasitemia or to resolve clinical

symptoms despite the use of an antimalarial drug at correct doses, may due to genetic factors. Furthermore, incomplete adherence (compliance), poor drug quality, incorrect dosing, compromised drug absorption, interactions with other drugs, vomiting of the medicine, and unusual pharmacokinetics are among other causes.

## Conclusion

Our results highlight the importance of blood cell count parameters in monitoring treatment course and outcome during antimalarial treatment. This result must be taken into account in the provision of medical and drug service deliveries, to improve the efficiency and reduce the costs of long-term hospital stays, as well as unfavorable outcomes, especially in low-income setting countries. Thus, antimalaria treatment failure is dangerous for the patients and for the community, as it may exacerbate malaria transmission and increase the emergence and spread of antimalarial drug resistance. More studies are needed to investigate the molecular and genetic factors related to antimalarial treatment failure.

## Supporting information

**S1 Data.**
(XLSX)

## Author Contributions

**Conceptualization:** Euclides Nenga Manuel Sacomboio, Eduardo Ekundi Valentim.

**Data curation:** Euclides Nenga Manuel Sacomboio, Ngiambudulu M. Francisco, Eduardo Ekundi Valentim.

**Formal analysis:** Euclides Nenga Manuel Sacomboio, Eduardo Ekundi Valentim.

**Funding acquisition:** Euclides Nenga Manuel Sacomboio.

**Investigation:** Euclides Nenga Manuel Sacomboio, Silvana Teresa da Costa Salvador, Joaquim António João, Eduardo Ekundi Valentim.

**Methodology:** Euclides Nenga Manuel Sacomboio, Eduardo Ekundi Valentim.

**Project administration:** Euclides Nenga Manuel Sacomboio.

**Resources:** Euclides Nenga Manuel Sacomboio, Silvana Teresa da Costa Salvador, Joaquim António João, Daisy Viviana Sebastião Bapolo, Eduardo Ekundi Valentim.

**Software:** Euclides Nenga Manuel Sacomboio.

**Supervision:** Euclides Nenga Manuel Sacomboio, Joana Morais, Eduardo Ekundi Valentim.

**Validation:** Euclides Nenga Manuel Sacomboio, Cruz dos Santos Sebastião, Silvana Teresa da Costa Salvador, Joaquim António João, Daisy Viviana Sebastião Bapolo, Ngiambudulu M. Francisco, Joana Morais, Eduardo Ekundi Valentim.

**Visualization:** Euclides Nenga Manuel Sacomboio, Cruz dos Santos Sebastião, Silvana Teresa da Costa Salvador, Joaquim António João, Daisy Viviana Sebastião Bapolo, Ngiambudulu M. Francisco, Joana Morais, Eduardo Ekundi Valentim.

**Writing – original draft:** Euclides Nenga Manuel Sacomboio, Joaquim António João, Daisy Viviana Sebastião Bapolo, Ngiambudulu M. Francisco, Joana Morais, Eduardo Ekundi Valentim.

**Writing – review & editing:** Euclides Nenga Manuel Sacomboio, Cruz dos Santos Sebastião, Joaquim António João, Daisy Viviana Sebastião Bapolo, Ngiambudulu M. Francisco, Joana Morais, Eduardo Ekundi Valentim.

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
