## [Decision Letter · Decision Letter 0]

16 Feb 2022

PONE-D-21-40976\\EVALUATION OF BLOOD CELL COUNT PARAMETERS AS PREDICTORS OF TREATMENT FAILURE OF MALARIA IN ANGOLA: AN OBSERVATIONAL STUDYPLOS ONE

Dear Dr. Sacomboio

Thank you for submitting your manuscript to PLOS ONE. After careful consideration, we feel that it has merit but does not fully meet PLOS ONE’s publication criteria as it currently stands. Therefore, we invite you to submit a revised version of the manuscript that addresses the points raised during the review process.

We look forward to receiving your revised manuscript.

Kind regards,

José Luiz Fernandes Vieira

Academic Editor

PLOS ONE

Journal Requirements:

a) Did participants provide their written or verbal informed consent to participate in this study?

**Comments to the Author**

1. Is the manuscript technically sound, and do the data support the conclusions?

Reviewer #1: Partly

Reviewer #2: Yes

2. Has the statistical analysis been performed appropriately and rigorously? 

Reviewer #1: Yes

Reviewer #2: Yes

3. Have the authors made all data underlying the findings in their manuscript fully available?

Reviewer #1: No

Reviewer #2: Yes

4. Is the manuscript presented in an intelligible fashion and written in standard English?

Reviewer #1: No

Reviewer #2: Yes

5. Review Comments to the Author

Reviewer #1: Dear authors,

First, I want to congratulate your study. However, after analysis, some questions must be answered or justified before publication

His study deals with “Evaluation of Blood Cell Count Parameters as Predictors of Malaria Treatment Failure in Angola: An Observation Study.

I. Abstract section: includes all the necessary information.

II- Background: Contains all necessary information.

III. Methodology: In the recruitment of the patient the authors cite the use of a questionnaire, I suggest that it be sent as supplementary material; laboratory procedure, why the authors do not include the following parameters: MCH, monocyte, eosinophil, NLR, MLR data? If the authors decide to include only these variables, a very careful assessment is required, since these values may not be adequate for a population with specific characteristics, considering their age groups and the geographic context in which they live. In addition, the nutritional status and the prevalence of intestinal parasites can influence these hematological indexes, and in malaria-endemic areas, it is imperative to consider these risk factors. Mainly due to the characteristics of patients included in the study, that is, males, students, and residents in peri-urban regions. I suggest to the authors to include or discuss these confounding factors. this bias.

IV. Results: In the evaluation of the erythrogram count, you only say that the patients had Hb volume below the reference values, however, the mean value was not mentioned. In a similar manner, other results should in described. The authors highlight that the resistance rate was higher in a group with low levels of Hb. A robust discussion on the role of the intraerythrocytic cycle of Plasmodium parasitism on the lysis of erythrocytes should be included as during the blood-stage, both P. Vivax and P. Falciparum can cause anemia. So, this result was not expected? Finally, what’s the novelty of the study?

Reviewer #2: The study study aimed to evaluate blood cell count parameters in the prediction of treatment failure in patients hospitalized for malaria. This is a good original paper and new ways of malaria and treatment failure had been proposed. However, there are some issues that need clarifications/revision.

Introduction

1. Paragraph should be re-arrange from "malaria" then "hematilogical alteration" and then "treatment failure"

2. There are several studies examing malaria and CBC as indicator for malaria infection. Can author review more studies and add them in the introduction?

3. Can authors review these little articles and cited them "little evidence exists whether alterations of blood cell count may predict malaria treatment failure that may lead to possible malaria drug resistance."

Methods

1. Selection criteria for participants should be explained separately in the subsection. What are the inclusion and exclusion criteria?

2. In the sentence "The study did not include mean corpuscular

hemoglobin (MCH), monocyte count, eosinophil count, neutrophil-lymphocyte ratio (NLR), and monocytelymphocyte ratio (MLR) due to appropriate devices to perform the counting of these parameters." What did "due to appropriate devices" mean? inporpiate?

3. In the sentence "Malaria diagnosis was confirmed by using rapid malaria antigen tests (SD-Bioline Malaria AG Pf/PAN) followed by microscopy technique of direct visualization of the parasite by Giemsa-stained peripheral

blood thickening [14]" RDT is a confirmatory or screening test for malaria? Please clarify.

4. In the statistical analysis, authors should mention what are the purpose of using chi-square or odds ratio aimed for. Please explain.

Results

1. Plasmodium falciparum must be italic.

2. Author must listed the cutoff for hematological parameters and also reference (s) such as what is the high hemoglobin or low hemoglobin.

Discussion

Discussion is well-writen.

Paragraph 1. Authors need to write the discussion not repeat the results. Also, not refer the " (Table 1)" in the discussion.

English editing and formatting are extensively required.

6. PLOS authors have the option to publish the peer review history of their article (what does this mean?). If published, this will include your full peer review and any attached files.

Reviewer #1: No

Reviewer #2: No

---

## [Author Response · Author response to Decision Letter 0]

2 Mar 2022

REBUTTAL LETTER

Journal Requirements:

a) Did participants provide their written or verbal informed consent to participate in this study?

Answers:

1. The situations referring to the term of free and clarified consent were added in the manuscript in the first paragraph of the methodology, we hope to have been able to respond to the editor's request.

2. Regarding the restrictions on data sharing, we added a paragraph in Cover Latter that explains the reasons, since this data is not stored in an electronic database, but in the physical records of patients, although our study was developed during the period in which the patients were hospitalized, today these data can be obtained from the clinical records of patients with malaria treated at the Josina Machel hospital between December 2018 and January 2020, with the authorization of the hospital's management, as they are exclusive access for medical teams and the Ministry of Health.

5. Review Comments to the Author

Reviewer #1: Dear authors,

First, I want to congratulate your study. However, after analysis, some questions must be answered or justified before publication

His study deals with “Evaluation of Blood Cell Count Parameters as Predictors of Malaria Treatment Failure in Angola: An Observation Study.

I. Abstract section: includes all the necessary information.

II- Background: Contains all necessary information.

III. Methodology: In the recruitment of the patient the authors cite the use of a questionnaire, I suggest that it be sent as supplementary material; laboratory procedure, why the authors do not include the following parameters: MCH, monocyte, eosinophil, NLR, MLR data? If the authors decide to include only these variables, a very careful assessment is required, since these values may not be adequate for a population with specific characteristics, considering their age groups and the geographic context in which they live. In addition, the nutritional status and the prevalence of intestinal parasites can influence these hematological indexes, and in malaria-endemic areas, it is imperative to consider these risk factors. Mainly due to the characteristics of patients included in the study, that is, males, students, and residents in peri-urban regions. I suggest to the authors to include or discuss these confounding factors. this bias.

Answer: We agree with the reviewer's proposal and therefore we include the information that the reviewer suggested in the methodology in the last paragraph before the statistical analysis.

IV. Results: In the evaluation of the erythrogram count, you only say that the patients had Hb volume below the reference values, however, the mean value was not mentioned. In a similar manner, other results should in described. The authors highlight that the resistance rate was higher in a group with low levels of Hb. A robust discussion on the role of the intraerythrocytic cycle of Plasmodium parasitism on the lysis of erythrocytes should be included as during the blood stage, both P. Vivax and P. Falciparum can cause anemia. So, this result was not expected? Finally, what’s the novelty of the study?

Answer: We agree with the reviewer's proposals and include the mean values of patients who presented values outside the established ranges as reference values in both the erythrogram and the leukogram. We also include some paragraphs in the discussion about the role of the intraerythrocytic cycle in Plasmodium parasitism. The novelty of the study is that the focus of the present study is not related to the impact of the parasite on the blood count of patients, but rather how changes in the blood count can indicate failures in the treatment of patients with malaria, so far very few studies have been found evaluating this. , even fewer studies were carried out in malaria-endemic countries, this can be extremely important to understand the resistance to treatment by the few antimalarials that still have some therapeutic efficacy.

Reviewer #2: The study aimed to evaluate blood cell count parameters in the prediction of treatment failure in patients hospitalized for malaria. This is a good original paper and new ways of malaria and treatment failure had been proposed. However, there are some issues that need clarifications/revision.

Introduction

1. Paragraph should be re-arrange from "malaria" then "hematilogical alteration" and then "treatment failure"

2. There are several studies examing malaria and CBC as indicator for malaria infection. Can author review more studies and add them in the introduction?

3. Can authors review these little articles and cited them "little evidence exists whether alterations of blood cell count may predict malaria treatment failure that may lead to possible malaria drug resistance."

Answer: All the reviewer's proposals were met, we reorganized the text starting with "malaria", then "hematological alteration" and then "treatment failure". We sought to improve our introduction by including other studies that examine malaria and CBC as an indicator of infection. Although we found very few studies that talk about blood cell counts and resistance to antimalarial treatment, we tried to improve our approach in the introduction.

Methods

1. Selection criteria for participants should be explained separately in the subsection. What are the inclusion and exclusion criteria?

2. In the sentence "The study did not include mean corpuscular

hemoglobin (MCH), monocyte count, eosinophil count, neutrophil-lymphocyte ratio (NLR), and monocytelymphocyte ratio (MLR) due to appropriate devices to perform the counting of these parameters." What did "due to appropriate devices" mean? inporpiate?

3. In the sentence "Malaria diagnosis was confirmed by using rapid malaria antigen tests (SD-Bioline Malaria AG Pf/PAN) followed by microscopy technique of direct visualization of the parasite by Giemsa-stained peripheral

blood thickening [14]" RDT is a confirmatory or screening test for malaria? Please clarify.

4. In the statistical analysis, authors should mention what are the purpose of using chi-square or odds ratio aimed for. Please explain.

Answers: We accepted the reviewer's suggestion, although we did not separate them with a subsection, we first grouped together the aspects that allowed the inclusion and later we explain the criteria that led to the exclusion of participants. We included in the methodology the reason why some blood count data were not included, since not all patients had these data because in some cases the devices were inappropriate for performing them, thus, only data that all patients presented is that were included in the study. We tried to make it clear in the methodology that the rapid test was a screening test and confirmation was made with Giemsa stain. In statistical analysis the information on why we use the chi-square or odds ratio.

Results

1. Plasmodium falciparum must be italic.

2. Author must listed the cutoff for hematological parameters and also reference (s) such as what is the high hemoglobin or low hemoglobin.

Answers: we agree and comply with the reviewer's proposal, we tried to put all texts of Plasmodium falciparum in italics. In the first paragraph of erythrogram and leukogram in the discussion we have the reference values of each of the parameters included in the study, which presuppose that all those in the ranges below or above are considered worrying, we believe that it must have gone unnoticed by the reviewer, however, we have included in this paragraph the mean values of blood cell counts among patients who presented values below the reference values universally established and used in Angola.

Discussion

Discussion is well-writen.

Paragraph 1. Authors need to write the discussion not repeat the results. Also, not refer the " (Table 1)" in the discussion.

English editing and formatting are extensively required.

Answers: we tried as far as possible to meet the reviewer's proposals in the discussion and in relation to English, we even sent a native speaker so he could help us correct the manuscript.

O.B.S: On behalf of all authors, we would like to thank the reviewers for the rigor and quality of the review, we realized that they raised extremely important issues that greatly improve the quality of the data presented in the submitted manuscript.

---

## [Decision Letter · Decision Letter 1]

13 Apr 2022

EVALUATION OF BLOOD CELL COUNT PARAMETERS AS PREDICTORS OF TREATMENT FAILURE OF MALARIA IN ANGOLA: AN OBSERVATIONAL STUDY

PONE-D-21-40976R1

Dear Dr. Euclides Nenga Manuel Sacomboio

We’re pleased to inform you that your manuscript has been judged scientifically suitable for publication and will be formally accepted for publication once it meets all outstanding technical requirements.

Kind regards,

José Luiz Fernandes Vieira

Academic Editor

PLOS ONE

Additional Editor Comments (optional):

Reviewers' comments:

Reviewer's Responses to Questions

**Comments to the Author**

1. If the authors have adequately addressed your comments raised in a previous round of review and you feel that this manuscript is now acceptable for publication, you may indicate that here to bypass the “Comments to the Author” section, enter your conflict of interest statement in the “Confidential to Editor” section, and submit your "Accept" recommendation.

Reviewer #1: (No Response)

Reviewer #2: All comments have been addressed

2. Is the manuscript technically sound, and do the data support the conclusions?

Reviewer #1: Yes

Reviewer #2: Yes

3. Has the statistical analysis been performed appropriately and rigorously? 

Reviewer #1: Yes

Reviewer #2: Yes

4. Have the authors made all data underlying the findings in their manuscript fully available?

Reviewer #1: Yes

Reviewer #2: Yes

5. Is the manuscript presented in an intelligible fashion and written in standard English?

Reviewer #1: Yes

Reviewer #2: Yes

6. Review Comments to the Author

Reviewer #1: Dear authors, all information has been properly reallocated and justified. I congratulate you for that.

Reviewer #2: Congratulation to authors for responses well to my comments. Before publication, please look carefully at the consistency of your manuscript. Authors are responsible for all contents in their manuscript.

7. PLOS authors have the option to publish the peer review history of their article (what does this mean?). If published, this will include your full peer review and any attached files.

Reviewer #1: No

Reviewer #2: No

---

## [Editor Report · Acceptance letter]

25 Apr 2022

PONE-D-21-40976R1 

EVALUATION OF BLOOD CELL COUNT PARAMETERS AS PREDICTORS OF TREATMENT FAILURE OF MALARIA IN ANGOLA: AN OBSERVATIONAL STUDY 

Dear Dr. Sacomboio:

I'm pleased to inform you that your manuscript has been deemed suitable for publication in PLOS ONE. Congratulations! Your manuscript is now with our production department. 

Kind regards, 

on behalf of

Dr. José Luiz Fernandes Vieira 

Academic Editor

PLOS ONE